# STAR-CAPS: Capsule Networks with Straight-Through Attentive Routing

**Karim Ahmed**
Department of Computer Science
Dartmouth College
karim@cs.dartmouth.edu

**Lorenzo Torresani**
Department of Computer Science
Dartmouth College
LT@dartmouth.edu

## Abstract

Capsule networks have been shown to be powerful models for image classification, thanks to their ability to represent and capture viewpoint variations of an object. However, the high computational complexity of capsule networks that stems from the recurrent dynamic routing poses a major drawback making their use for large-scale image classification challenging. In this work, we propose STAR-CAPS a capsule-based network that exploits a *straight-through attentive routing* to address the drawbacks of capsule networks. By utilizing *attention* modules augmented by *differentiable binary routers*, the proposed mechanism estimates the routing coefficients between capsules without recurrence, as opposed to prior related work. Subsequently, the routers utilize *straight-through estimators* to make binary decisions to either *connect* or *disconnect* the route between capsules, allowing stable and faster performance. The experiments conducted on several image classification datasets, including MNIST, SmallNorb, CIFAR-10, CIFAR-100 and ImageNet show that STAR-CAPS outperforms the baseline capsule networks.

## 1   Introduction

Convolutional neural networks (CNNs) have achieved successful performance on different computer vision tasks [7, 14, 25, 6, 22]. By using local receptive fields and shared weights, CNNs can identify the existence of entities regardless of their spatial locations (translation invariance). CNNs use a deep sequence of convolutional layers or max pooling operations which downsample the spatial size. Max-pooling is considered a primitive form of routing in which the output only attends to the most active neuron in the pool. By throwing away information about the precise position of an entity, max-pooling achieves some translation invariance. To mitigate the viewpoint variations of an entity, CNNs combine the activities of the pool, i.e., overlapping the sub-sampling pools. However, CNNs fail to represent the part-whole relationships of the entities, thus they cannot detect radically new viewpoints due to losing the precise spatial relationships in the max-pooling operations. Contrarily, capsule networks [23, 8] utilize trainable viewpoint-invariant transformations that learn to represent part-whole relationship of the entities. Although, capsule models have been shown to be powerful models to detect viewpoint variations compared to the traditional convolutional neural networks [23, 8], the computational complexity of these models during training and inference is a major drawback which limits utilizing these networks efficiently on large-scale image classification datasets. This poses a dilemma: choosing between capsule networks and convolutional neural networks requires sacrificing either the computational efficiency or the mechanism to detect viewpoint variations, respectively. In this work, we present STAR-CAPS, a capsule-based architecture that utilizes a *straight-through attentive routing* to address the drawbacks of the recurrent dynamic routing. The proposed routing mechanism is based on efficient *attention* modules augmented by *differentiable binary routers*, which make routing decisions utilizing a set of straight-through gradient estimators [10, 1]. We outline the motivation and the contributions of our work, next.

## 1.1 Motivation and Contributions

The *computational complexity* of the capsule networks during the training stage as well as the inference, stems from the complex mechanisms of the voting and the routing steps. In the voting step, the lower-level $n$ capsules cast votes for the higher-level $m$ capsules. This is achieved by transforming the lower-level pose matrices using distinct $(n \times m)$ transformation matrices. For a capsule layer with kernel size of $k$, the voting step in one forward pass requires $(k^2 \times n \times m)$ matrix-matrix multiplications. In the routing step, the recurrent dynamic routing algorithms [8, 23] depend on multiple iterations to update the agreements. Each iteration requires additional expensive operations such as matrix multiplications or exponential functions. The routing complexity gets intensified in the EM routing algorithm [8] that requires two steps (E-step and M-Step) per iteration. Though a capsule network architecture has a fixed number of parameters, training and inference time can increase dramatically according to the number of routing iterations defined a priori as a hyperparameter.

To address the *computational complexity* of capsule networks, we replace the recurrent dynamic routing by a *non-recursive attention-based routing* mechanism. The motivation of our routing mechanism stems from the relation between the non-recurrent self-attention employed in the Transformer [26] and the recurrent dynamic routing [8, 23]. Compared to the recurrent neural networks, the self-attention [26] has been shown to be faster and more powerful. In fact, the recurrent dynamic routing can been seen as an attention mechanism, but in the opposite direction [8]. As an additional advantage of our proposed routing mechanism, the capsule network avoids the *underfitting/overfitting* caused by the improper setting of the number of routing iterations [23]. The experiments conducted by Sabour et al. [23] showed that fewer routing iterations may lead to underfitting, whereas large number of iterations cause overfitting; thus, training a capsule network often require trial and error to identify a satisfactory number of routing iterations for a specific task and dataset. Furthermore, compared to the baseline capsule network [8], our approach shows a stable and better performance without being sensitive to the predefined number of capsules in each layer and their initializations.

Our *main contributions* can be summarized as follows.

- To enable faster training and inference, we replace the recurrent dynamic routing mechanism by efficient attention modules augmented by differentiable binary routers, which exploit a group of straight-through gradient estimators to make routing decisions.
- As an additional benefit of the proposed routing mechanism, the capsule network avoids the underfitting/overfitting that occurs in the recurrent dynamic routing mechanisms, caused by choosing an improper number of iterations. Furthermore, our approach allows more stable performance without being sensitive to the predefined number of capsules and their initializations.
- We conducted different experiments on several image classification datasets, including MNIST, SmallNorb, CIFAR-10, CIFAR-100 and ImageNet. Our results show that STAR-CAPS outperforms the baseline capsule networks.

## 2 Background

### 2.1 Capsule Networks

A capsule neural network consists of capsule layers, where each layer is constructed from a set of capsules. A capsule is a unit that represents a group of neurons formulated as a vector [23] or a matrix [8] that reflects properties of an entity such as pose. Figure 1 shows traditional neural layers vs. capsule layers. In traditional neural networks, the neurons are connected through a set of weights learned during training. In capsule networks, the information flow between the lower-level and the higher-level capsules can be described in two steps: (1) *voting*, in which lower-level capsules cast votes for the higher-level capsules, and (2) *routing*, where lower-level and higher-level capsules are connected via routing coefficients learned by a dynamic routing algorithm. In DynamicCaps [23] the capsule is a vector that represents the pose, and its length indicates the existence of an entity. In EMCaps [8] the capsule has a pose matrix, and an activation scalar.

In general, the architecture of capsule networks [8, 23] consists of: (i) a traditional convolutional layer, (ii) a PrimaryCaps, a special convolutional capsule layer that converts activities into vector capsules [23] or matrix capsules [8], (iii) a set of convolutional capsule layers (ConvCaps layers) that learn the part-whole relationships of entities, (iv) the final capsule layer is ClassCaps which outputs the final class predictions. During *voting*, the pose of a lower-level capsule is multiplied by trainable weights (transformation matrix) to cast a vote for each higher-level capsule. Capsules

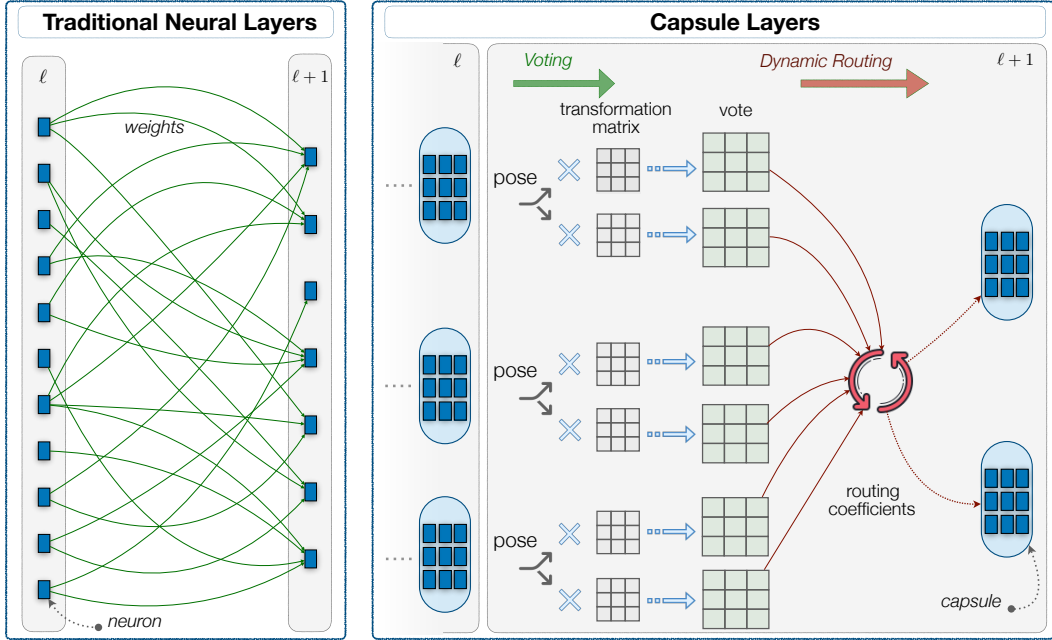

Figure 1: Traditional Neural Layers (left) vs. Capsule Neural Layers (right).

make use of this underlying linearity to allow learning and representing the part-whole relationships of the entities, thus detecting the viewpoint variations [8]. *Recurrent dynamic routing* is a routing-by-agreement iterative approach, in which each lower-level capsule sends its vote to the capsules in the higher level that agree. These agreements are achieved through many iterations of adjusting the routing coefficients. The routing-by-agreement algorithm in `DynamicCaps` [23] is a dynamic iterative mechanism based on coordinate descent optimization; whereas in `EMCaps` [8] the routing is based on an Expectation-Maximization procedure.

## 2.2 Attentions

The Transformer [26] relies on multi-head self-attentions to capture the dependencies between the input and the output. The self-attention layers decide how to attend various parts of the input and generates attention coefficients to update the representations. Compared to the recurrent layers used in recurrent neural networks, the self-attention layers that do not use any recurrence have been shown to be faster and more powerful [26]. It can be noticed the relation between the self-attention mechanism employed in the Transformer [26] and the recurrent dynamic routing approaches [8, 23] in capsule networks. Dynamic routing [8, 23] can been seen as an attention mechanism, but in the opposite direction. The dynamic routing is a bottom-up approach where the competition is between the higher-level capsules that a lower-level capsule might send its vote to; whereas the attention-based routing is a top-down approach where the competition is between the lower-level capsules that a higher-level capsule might attend to. Several prior work have utilized attention mechanisms with capsule-based networks. Zhang et al. [30] proposed a relation extraction approach based on capsule networks with attention; however, the proposed attention mechanism was used as an augmentation to a capsule network [23] that utilizes a dynamic routing mechanism. Li et al. [18] proposed to improve the information aggregation for multi-head attention with a dynamic routing algorithm. Xinyi et al. [29] proposed a capsule graph network that utilizes an attention module to scale node embeddings followed by dynamic routing to generate graph capsules. Differently from the prior work, we propose a capsule-based architecture that replaces the recurrent dynamic routing mechanism by a non-recurrent attentive routing mechanism.

## 2.3 Straight-through Estimators

Our approach utilizes routing modules to make binary decisions to either connect or disconnect the route between capsules. Propagating gradients through discrete stochastic nodes has been studied in the literature, for instance Bengio et al. [1] proposed a straight-through estimator to estimate and propagate the gradients through discrete stochastic neurons. In STAR-CAPS, we adopt a straight-

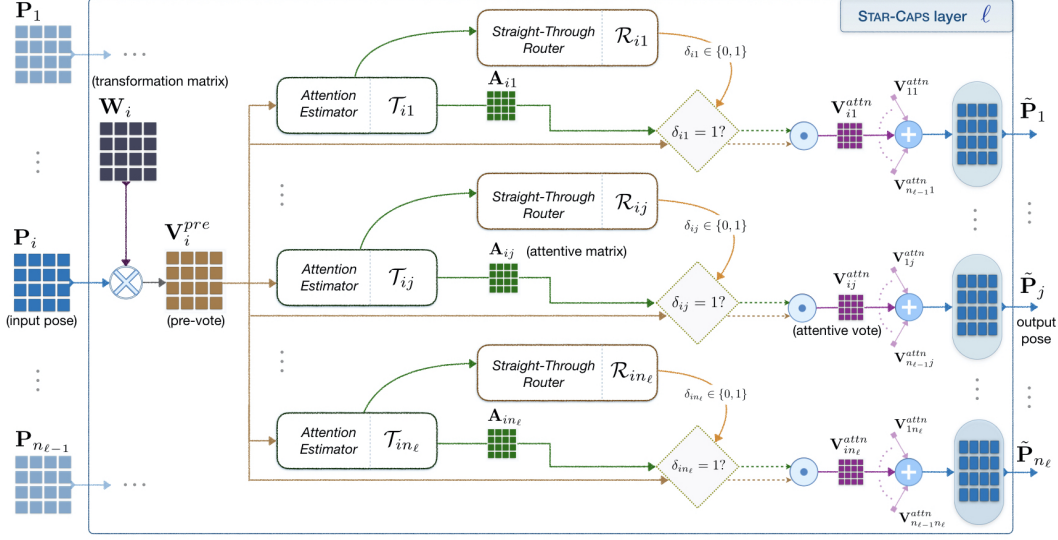

Figure 2: Overview of a STAR-CAPS layer.

through estimator based on Gumbel-Softmax [10] to implement the binary routers. Differently from our approach, Guo et al. [5] and Viet et al. [27] uses Gumbel-Softmax [10], to decide which layers in a CNN to fine-tune during transfer learning, and for adaptive inference in CNNs, respectively.

# 3 STAR-CAPS Architecture

STAR-CAPS is a capsule-based network that utilizes a *straight-through attentive routing* mechanism. We opt to formulate each capsule as a matrix rather than a vector to save parameters [8]. Given the pose features from the lower-level capsules, we transform the pose through shared trainable weight matrices, i.e. a single weight matrix between each lower-level capsule and all the higher-level capsules. We call the output of this transformation the *pre-vote*. The routing between the lower-level and higher-level capsules takes place through two components: the *Attention Estimator* and the *Straight-through Router*. Given the pre-vote, each Attention Estimator estimates an attentive coefficient matrix that acts as a soft relevance signal for each higher-level capsule. Additionally, each Attention Estimator is sequentially augmented by a Straight-through Router, a differentiable binary router that acts as a *gate*. This router estimates a binary signal that decides whether to connect or disconnect the current route between the lower-level capsule and the higher-level capsule. The binary signal estimated by the router can be seen as a hard-attention coefficient, albeit differentiable. Conceptually, each route can be seen as a double-attention (soft & hard) mechanism. Between each lower-level capsule and all the higher-level capsules, we build a tree of double-attentions; thus, creating a forest of double-attentions in each capsule layer. During training, each double-attention component learns the connectivity between capsules in a stochastic dynamic manner, yet differentiable, which can be a seen as an attention-based connectivity search mechanism. Next, we give an overview of the overall architecture (§ **3.1**), then we discuss the Attention Estimator (§ **3.2**) and the Straight-through Router (§ **3.3**).

## 3.1 Overview

Our architecture starts with a regular convolutional layer (Conv) with kernel ($\breve{k} \times \breve{k}$), $\breve{c}$ channels and ReLU non-linearity, followed by a sequence of capsule layers. The first capsule layer is a primary capsule type (PrimaryCaps) [8], followed by a set of $m$ convolutional capsule type (ConvCaps). PrimaryCaps and ConvCaps layers have kernel size of ($k \times k$). The final layer (ClassCaps) predicts the classes, where each class is represented by one capsule, i.e. the number of capsules is equal to the number of classes. Each capsule layer $\ell \in \{0, \dots, m, m+1\}$ contains $n_\ell$ capsules. Each capsule is composed of a *pose matrix* defined explicitly, whereas the *activation* is implicitly encoded as we will discuss later. We use the following notation to define a capsule network:

$$\big\{ \texttt{Conv}(\breve{k}, \breve{c}), \texttt{PrimaryCaps}(k, n_0), \{\texttt{ConvCaps}_\ell(k, n_\ell) \mid 1 \le \ell \le m\}, \texttt{ClassCaps}(n_{m+1}) \big\}$$

`ConvCaps` is the key layer of the architecture where the routing between capsules takes place. Figure 2 illustrates an overview of the (`ConvCaps` $\ell$) using our proposed routing mechanism. The input of `ConvCaps` $\ell$ is the set of the pose matrices $\mathbb{P}_{\ell-1} = \big\{ \mathbf{P}_i \in \mathbb{R}^{p \times p} \mid i \in \{1, \ldots, n_{\ell-1}\} \big\}$ generated by the lower-level capsules in layer $\ell - 1$. Correspondingly, the output is the set of pose matrices $\mathbb{P}_\ell = \big\{ \tilde{\mathbf{P}}_j \in \mathbb{R}^{p \times p} \mid j \in \{1, \ldots, n_\ell\} \big\}$ generated by the higher-level capsules defined in the current layer $\ell$. Pose matrices are not stored parameters and they act as a group of activities.

***Transformation of Input Pose:*** Given $\mathbb{P}_{\ell-1}$, each input *pose matrix* [1] $\mathbf{P}_i \in \mathbb{R}^{p \times p}$ is multiplied by a trainable transformation matrix $\mathbf{W}_i \in \mathbb{R}^{p \times p}$. We point out that the output of each transformation is not the actual vote considering that there is a single transformation matrix for each input pose matrix. Thus, we call the transformed pose, the *pre-vote* $\mathbf{V}_i^{pre} \in \mathbb{R}^{p \times p}$.

$$\mathbf{V}_i^{pre} = \mathbf{P}_i \mathbf{W}_i, \quad \forall i \in \{1, \ldots, n_{\ell-1}\} \tag{1}$$

***Attentions:*** For capsule $i$, we build a tree structure of *Attention Estimator* (§ **3.2**) modules. Each module estimates distinct *attentive matrix* $\mathbf{A}_{ij}$ for every capsule $j$, given the shared $\mathbf{V}_i^{pre}$

$$\big\{ \mathcal{T}_{ij} : \quad \mathbf{V}_i^{pre} \in \mathbb{R}^{p \times p} \to \mathbf{A}_{ij} \in \mathbb{R}^{p \times p} \mid i \in \{1, \ldots, n_{\ell-1}\}, \ j \in \{1, \ldots, n_\ell\} \big\} \tag{2}$$

***Routers:*** Given the *attentive matrix* $\mathbf{A}_{ij}$ estimated by $\mathcal{T}_{ij}$ (Eqn. 2), a *Straight-Through Router* (§ **3.3**) $\mathcal{R}_{ij}$ acting as a gate, estimates a binary decision value $\delta_{ij} \in \{0, 1\}$ indicating whether to *disconnect* ($\delta_{ij} = 0$) or *connect* ($\delta_{ij} = 1$) the route between capsules $i$ and $j$. This mechanism can be seen as a hard attention, yet differentiable (see (§ **3.3**)), where each $\mathcal{R}_{ij}$ sends its hard attention signal to the higher-level capsules.

$$\big\{ \mathcal{R}_{ij} : \quad \mathbf{A}_{ij} \in \mathbb{R}^{p \times p} \to \delta_{ij} \in \{0, 1\} \mid i \in \{1, \ldots, n_{\ell-1}\}, \ j \in \{1, \ldots, n_\ell\} \big\} \tag{3}$$

***Calculation of Output Pose:*** Each higher-level capsule $j$, receives a list of $n_{\ell-1}$ tuples of features, each tuple $(\mathbf{V}_i^{pre}, \mathbf{A}_{ij}, \delta_{ij})$ is generated by the lower-level capsule $i$. The output pose matrix $\tilde{\mathbf{P}}_j \in \mathbb{R}^{p \times p}$ of capsule $j$ in `ConvCaps` $\ell$ is calculated as follows:

$$\tilde{\mathbf{A}}_{ij} = \mathbf{A}_{ij} \oslash \sum_{\substack{i=1 \\ \delta_{ij}=1}}^{n_{\ell-1}} \mathbf{A}_{ij} \quad ; \quad \tilde{\mathbf{P}}_j = \sum_{\substack{i=1 \\ \delta_{ij}=1}}^{n_{\ell-1}} \mathbf{V}_i^{pre} \odot \tilde{\mathbf{A}}_{ij} \tag{4}$$

$\oslash$ is element-wise division, $\sum_{\delta_{ij}=1}$ is a summation masked by $\delta_{ij}$, $\odot$ is element-wise product, and $(\mathbf{V}_i^{pre} \odot \tilde{\mathbf{A}}_{ij})$ is the *attentive vote* $\mathbf{V}_{ij}^{attn}$

***Activation Probablity:*** The `ClassCaps` layer ($\ell = m + 1$) outputs the final predictions, where each capsule represents a single class. The activation probability ($a_t$) indicates the presence of an object class $t$. This activation is implicitly encoded in the capsule. Given $\tilde{P}_t$, we calculate $a_t$ as follows:

$$a_t = \mathcal{M}\Big( \sigma(\tilde{P}_t) \Big) = \frac{1}{p^2} \sum_{s=1}^{p} \sum_{\hat{s}=1}^{p} \sigma(\tilde{P}_t[s, \hat{s}]), \quad t \in \{1, \ldots, n_{m+1}\} \tag{5}$$

$\sigma$ is a sigmoid function, $\mathcal{M}$ is a *global average pooling* [19].

***Loss Function:*** Given the activations ($a_t$), $t \in \{1, \ldots, n_{m+1}\}$, we calculate the *spread loss* [8].

## 3.2   Attention Estimator

The role of the *Attention Estimator* ($\mathcal{T}_{ij}$) (Eqn.2) is to estimate the *attentive matrix* $\mathbf{A}_{ij} \in \mathbb{R}^{p \times p}$ with $c$ channels. We propose an efficient bottleneck architecture which consists of 3 convolutional layers. The architecture [2] starts with `Conv2D`($c$, 1x1, $d$) and ends with `Conv2D`($d$, 1x1, $c$), followed by a BatchNorm [9] and a LeakyRelu [20]. We set $c = k^2$ and $d \le k^2$. Inspired by the recent work of Wu et al. [28], we design the middle layer as a *lightweight 2D convolution* (`LightConv2D`) with $H$ attention heads, which is a depth-wise separable [2, 11, 24] convolution that shares $\frac{d}{H}$ output channels, and the weights are normalized using a `Softmax2D`.

### 3.3 Straight-Through Router

Given the attentive matrix $\mathbf{A}_{ij}$, the *Straight-Through Router* ($\mathcal{R}_{ij}$) (Eqn.3) estimates a binary decision signal $\delta_{ij} \in \{\mathbf{0}: disconnect, \mathbf{1}: connect\}$. We design the router to be a differentiable hard attention module. The intuition is to allow learning the *attention-based connectivity* or *relevance* between capsules. The *Straight-Through Router* consists of two sequential sub-modules, *Decision-Learner* and *Decision-Maker*. We discuss the details [3] next.

***Decision-Learner:*** The *Decision-Learner* learns a pair of decision scores $\mathbf{\Pi} \in \mathbb{R}^2$, we will assume $\mathbf{\Pi} = \{\pi_0, \pi_1\}$. Conceptually, it can be defined as $\mathcal{DL}_{\theta_{\mathcal{DL}}} : \mathbf{A} \in \mathbb{R}^{c \times p \times p} \longrightarrow \mathbf{\Pi} \in \mathbb{R}^2$. First, we apply a global average pooling [19] on $\mathbf{A}$, to capture the confidence maps [19] of the $c$ channels and to reduce the computational complexity. Then, we apply $\texttt{Conv2D}(c, 1x1, c)$ followed by a BatchNorm [9] and a LeakyRelu [20]. Finally, we apply $\texttt{Conv2D}(c, 1x1, 2)$ to generate unnormalized decision real-valued scores $\mathbf{\Pi}$. Empirically, this simple architecture enables fast and efficient estimation of the decision scores, which is essential to minimize the overall computational overhead of the routing process between the capsules.

***Decision-Maker:*** Given the real-valued scores $\mathbf{\Pi}$, we estimate a binary decision parameter $\delta \in \{0, 1\}$ that indicates a decision chosen from a set of two mutually exclusive and exhaustive events, (i) *connect* (if $\delta = 1$) or (ii) *disconnect* (if $\delta = 0$) the route between the current two capsules. The *Decision-Maker* can be represented as $\mathcal{DM} : \mathbf{\Pi} \in \mathbb{R}^2 \longrightarrow \mathbf{I} \sim Bernoulli(\delta)$, where $\mathbf{I}$ is a bernoulli (indicator) random variable parameterized by $\delta \in \{0, 1\}$. Conceptually, this representation can be seen as a binarization function of the real-valued scores $\mathbf{\Pi}$ such that each value in the pair of the binary outcomes is the complement of the other. A simple way to implement $\mathcal{DM}$, is to adopt a deterministic approach during training such as selecting the position with the maximum value of $\mathbf{\Pi}$. However, this approach is not differentiable and tends to memorize the same generated binary samples throughout training. Propagating gradients through discrete stochastic nodes has been studied in the literature, for example Bengio et al. [1], proposed a "straight-through estimator" to estimate and propagate the gradients through discrete stochastic neurons. In our work, we adopt a "straight-through estimator" based on Gumbel-Softmax [10].

Given a discrete categorical distribution with classes probabilities, we can draw samples using the Gumble-Max trick [21, 4]. In our case, we have two classes (*disconnect* and *connect*), and we assume that the unnormalized real-valued scores $\{\pi_0, \pi_1\}$ generated by $\mathcal{DL}_{\theta_{\mathcal{DL}}}$ are the log probabilities of these two classes, i.e. $\pi_\kappa = log[p_\kappa]$ where $\kappa \in \{0, 1\}$ and $p_\kappa$ is the probability of class $\kappa$. Thus, we can draw a sample from a Bernoulli distribution (as a special case of the categorical distribution) parameterized by $\{p_0, p_1\}$ as follows:

$$\mu = \texttt{argmax}_{\{0, 1\}}\big[(\pi_0 + \mathbf{g}_0), (\pi_1 + \mathbf{g}_1)\big] \tag{6}$$

where $\{\mathbf{g}_0, \mathbf{g}_1\}$ are i.i.d samples drawn from the Gumbel distribution *Gumbel*(0, 1) acting as a noise to introduce stochasticity, *Gumbel*(0, 1) is defined as $-log(-log(U))$, $\mathrm{U} \sim Uniform(0, 1)$. The $\texttt{argmax}$ is non-differentiable, however. Alternatively, we can use the Gumbel-Softmax Estimator [10] to sample from a discrete Bernoulli distribution, by using a $\texttt{softmax}$ as a continuous differentiable approximation to $\texttt{argmax}$.

$$\nu_\kappa = \frac{\exp(\pi_\kappa + \mathbf{g}_\kappa)/\tau}{\sum_{\hat{\kappa}=0}^{1} \exp(\pi_{\hat{\kappa}} + \mathbf{g}_{\hat{\kappa}})/\tau}, \quad \kappa \in \{0, 1\}, \ \tau \text{ is the temperature} \tag{7}$$

The *Decision-Maker* $\mathcal{DM}$ is implemented as a *straight-through Gumbel-softmax* [10], which uses Eqn.(6) in the *forward* pass. Thus, the binary decision parameter $\delta = \mu$. In the *backward* pass, the gradients of the binary samples are approximated by computing the gradients of the continuous $\texttt{softmax}$ Eqn.(7), i.e. $\nabla_\theta \mu \approx \nabla_\theta \nu$.

## 4 Experiments

We evaluated our approach on the task of image classification using the following datasets: MNIST [15], SmallNorb [16], CIFAR-10 [13], CIFAR-100 [13], and ImageNet [3]. The baseline models are based on $\texttt{EMCaps}$ [8], since the capsule in $\texttt{EMCaps}$ [8] is formulated as a matrix similar to our approach, and it showed better general performance compared to $\texttt{DynamicCaps}$ [23].

***Models and training settings.*** Unless otherwise specified, STAR-CAPS models as well as `EMCaps` [8] models consist of a $(5 \times 5)$ `Conv` with ReLU, 1 `PrimaryCaps`, 2 `ConvCaps`, and 1 `ClassCaps`. The kernel size of `ConvCaps` is $k = 3$. The number of channels of `Conv` and the number of capsules in each layer will be specified for each model using the following notation: #capsules=$\{\check{c}, n_0, n_1, n_2, n_3\}$ as described in (§ **3**). We use Adam [12] optimizer, with coefficients $(0.9, 0.999)$. The initial learning rate is $0.01$, and the training batch size $T = 128$.

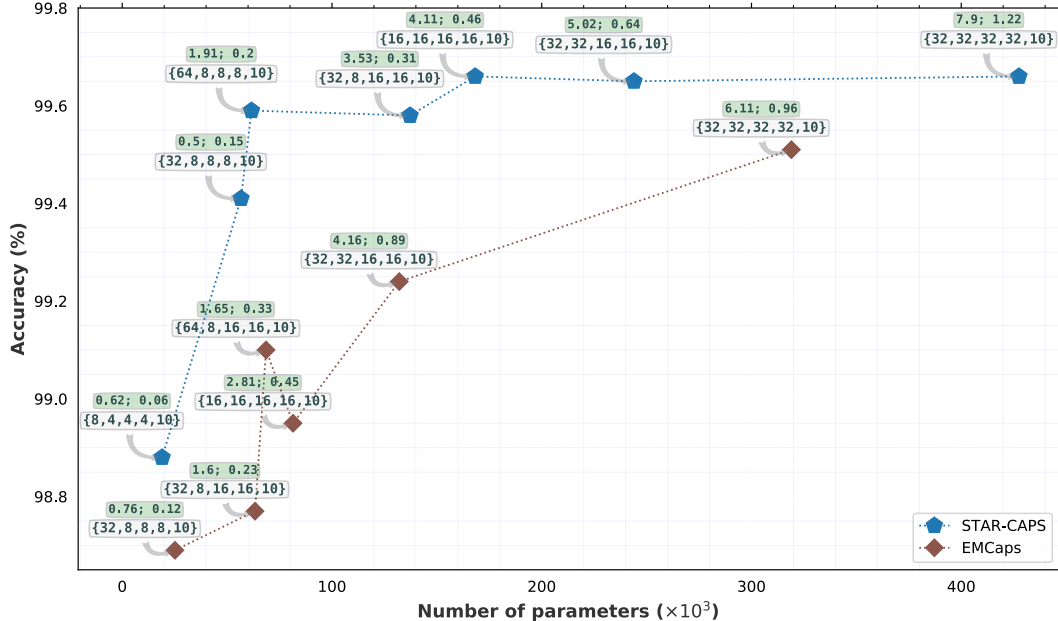

Figure 3: Comparison between STAR-CAPS and `EMCaps` [8] models trained on MNIST. The gray box shows #capsules $\{\check{c}, n_0, n_1, n_2, n_3\}$; whereas the green box shows the (training time; testing time) in secs per batch.

## 4.1 Evaluation on MNIST

We perform training on MNIST [15] gray-level 28x28 images. The dataset consists of 60K training images and 10K testing images. We compare different STAR-CAPS and `EMCaps` models in terms of accuracy, training time, and testing time. For STAR-CAPS models, we set $\hat{k} = 3$, and $d = 3$. For `EMCaps` models, the number of routing iterations is 2. Figure 3 shows the classification accuracy of different STAR-CAPS and `EMCaps` models. Each model varies in terms of the number of capsules and the number of parameters. We notice that STAR-CAPS models yield better accuracy compared to `EMCaps` models. Furthermore, STAR-CAPS shows more stable performance and faster training and testing time. We point out that we could not train an `EMCaps` model with larger number of parameters than the model shown in Figure 3, i.e. the `EMCaps`:$\{32, 32, 32, 32, 10\}$ and 319K parameters. This is because larger `EMCaps` models, in addition for being very expensive to train, they were overfitting under different hyperparameters settings.

Table 1: Performance sensitivity to the predefined # capsules: STAR-CAPS vs. `EMCaps` evaluated on MNIST. We report (mean±std) of the test accuracy of 3 runs.

| Model | #Params | Accuracy(mean±std) |
|---|---|---|
| STAR-CAPS:$\{32, 4, 64, 4, 10\}$ | 143K | 99.49±0.11 |
| EMCaps:$\{32, 4, 64, 4, 10\}$ | 77K | 96.89±0.13 |
| STAR-CAPS:$\{64, 8, 64, 8, 10\}$ | 281K | 99.57±0.09 |
| EMCaps:$\{64, 8, 64, 8, 10\}$ | 159K | 98.12±0.12 |

Our experiments show that the performance of the baseline `EMCaps` [8] models can be sensitive to the numbers of capsules defined for each layer and their initializations. For instance, on MNIST, training an `EMCaps` model in which one or more capsule layer contain a large number of capsules, and the lower-level or the higher-level capsule layers have small number of capsules, the performance of this model becomes unstable even with careful initializations of the capsules. On the other hand, STAR-CAPS mitigates this problem by learning to disconnect the superfluous capsules during routing

more efficiently. In Table 1, we compare the performance of STAR-CAPS and EMCaps using two model variations that use different number of capsules.

## 4.2 Evaluation on SmallNorb

SmallNorb [16] contains gray-level stereo images of 5 toy classes. Each image represents 18 azimuths (range 0-340), 6 lightning variations, and 9 elevations. We follow the data preprocessing as in EMCaps [8], yielding randomly cropped training image patches of size 32x32. We compare the performance of two different STAR-CAPS and EMCaps models with comparable number of parameters. STAR-CAPS:$\{32, 8, 8, 8, 5\}$ achieves 98.0% compared to EMCaps:$\{64, 8, 16, 16, 5\}$ that achieves 97.8%; whereas both STAR-CAPS:$\{32, 32, 16, 16, 5\}$ and EMCaps:$\{32, 32, 32, 32, 5\}$ achieve 98.2%.

Table 2: Detection of novel viewpoints on SmallNorb

| | Type1: (low capacity) | | Type2: (high capacity) | | |
| --- | --- | --- | --- | --- | --- |
| Model | EMCaps | STAR-CAPS | CNN | EMCaps | STAR-CAPS |
| #Params | 68K | 73K | 4.2M | 316K | 318K |
| Familiar | 95.66±0.03 | 95.72±0.02 | 96.3 | 96.3 | 96.3 |
| Novel | 86.12±0.05 | 86.07±0.03 | 80.0 | 86.5 | 86.3 |

***Detection of novel viewpoints:*** We use SmallNorb to evaluate the ability of STAR-CAPS to detect novel viewpoints, similar to the experiments in EMCaps [8]. We create a subset of SmallNorb with two parts, each part contains images of distinct azimuths range as follows: "Train-viewpoints" which contains the training images with azimuths (300, 320, 340, 0, 20, 40), and "Test-viewpoints" that has the testing images of azimuths range (60-280). We train two types of models (low capacity and high capacity) for STAR-CAPS and EMCaps on "Train-viewpoints", and we evaluate the models on "Test-viewpoints". Table 2 shows two types of experiments on SmallNorb (novel, familiar viewpoints). *Type1:* 3 runs of EMCaps:$\{64, 8, 16, 16, 5\}$, STAR-CAPS:$\{32, 8, 8, 8, 5\}$, fully trained on familiar views and tested on both novel and familiar views. *Type2:* EMCaps:$\{32, 32, 32, 32, 5\}$, STAR-CAPS:$\{32, 32, 16, 16, 5\}$, trained on familiar views and early stopped when test accuracy reached 96.3% (as the CNN model in [8]). In *Type1*, we notice that STAR-CAPS achieves comparable results (small difference in accuracy) to EMCaps both on familiar and novel viewpoints. In *Type2*, on the novel viewpoints, STAR-CAPS performs dramatically better than CNN model (+6.3%) and its accuracy is only slightly lower than EMCaps (-0.2%).

## 4.3 Evaluation on CIFAR-10/CIFAR-100

CIFAR-10 [13] and CIFAR-100 [13] datasets contain images of size 32x32, with 10 classes and 100 classes, respectively. For each dataset, the training set consists 50,000 images, and the testing set has 10,000 images. We train a CIFAR-10 model based on STAR-CAPS:$\{32, 8, 8, 8, 10\}$, which achieves a test accuracy of 91.23% with test time of 0.21 secs/batch, compared to EMCaps:$\{256, 32, 32, 32, 10\}$ that achieves 88.10%. Another relevant work, is the EncapNet [17] which achieves an accuracy of 88.07%. On CIFAR-100, our STAR-CAPS model achieves 67.66%, while an EMCaps:$\{256, 32, 32, 32, 100\}$ was not able to converge yielding 19%.

## 4.4 Evaluation on ImageNet

ImageNet [3] is a large-scale dataset with 1000 classes. As per our knowledge, there is no related work that was able to train EMCaps [8] model on ImageNet. We point out that EncapNet [17] model that reported preliminary results on ImageNet, was built upon a deep residual network [7] augmented by a capsule module. We construct a STAR-CAPS model that starts with 7x7 Conv layer and output 64 channels, followed by a single bottleneck residual block with 256 output channels. Afterwards, we add 4 capsule layers with 64 capsules for PrimaryCaps and 128 capsules for ConvCaps layers. The Top-1 validation accuracy of this model is 60.07% and the Top-5 accuracy is 85.66%.

## 5 Conclusion

We presented STAR-CAPS, a capsule-based network that utilizes a *straight-through attentive routing* to address the computational complexity of capsule networks. The proposed routing is a double-attention mechanism utilizing (a) *Attention Estimators* that estimate attention matrices between capsules, and (b) *Straight-Through Routers* to make binary connectivity decisions between capsules. Our experiments showed that STAR-CAPS outperforms the baseline capsule models.

**Acknowledgments**

This work was funded in part by NSF award CNS-120552. We gratefully acknowledge NVIDIA and Facebook for the donation of GPUs used for portions of this work.

## Footnotes

[1] For each input sample in the training batch, the size of the *pose matrix* is ($c \times p \times p$), where $c$ is the number of channels. For simplicity, we frequently omit $c$ from our notation.

[2] `Conv2D`($c$, 1x1, $d$) is a 2D convolution with $c$ input channels, 1x1 kernel size, $d$ output channels.

[3]Henceforth, for simplicity we omit the subscript index $(ij)$ from our notation.

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
