[Supplementary Material]

# - Supplementary Material -
# STAR-CAPS: Capsule Networks with Straight-Through Attentive Routing

**Karim Ahmed**
Department of Computer Science
Dartmouth College
karim@cs.dartmouth.edu

**Lorenzo Torresani**
Department of Computer Science
Dartmouth College
LT@dartmouth.edu

## A.    Additional Experiments

**affnist dataset**    We trained STAR-CAPS $\{32, 8, 16, 16, 10\}$ on MNIST following the data augmentation as in EMCaps [2]. The test accuracy of STAR-CAPS on affNIST [1] is 93.03% vs. 93.1% for EMCaps $\{32, 32, 32, 32, 10\}$.

**Performance of STAR-CAPS vs. CNNs**    Although the main purpose of STAR-CAPS is to alleviate the computational complexity of baseline capsule networks while being able to detect viewpoint variations, STAR-CAPS models achieve accuracies nearly on par with those modern CNN models. On CIFAR10, STAR-CAPS: 91.23%, #params=80K vs. ResNet20: 91.25%, #params=270K vs. ResNet110: 93.57%, #params=1.7M. On CIFAR100, STAR-CAPS: 67.66% vs. ResNet38: 68.54% vs. ResNet110: 71.21%. It is possible that scaling up STAR-CAPS models to match #params in ResNet, would lead to better performance.

**STAR-CAPS without ST-Router**    Removing ST-Router leads to lower performance. On MNIST, STAR-CAPS model $\{32, 8, 16, 16, 10\}$ achieves 99.41% without ST-Router and 99.59% with ST-Router; whereas STAR-CAPS $\{32, 4, 64, 4, 10\}$ achieves 98.37% without ST-Router and 99.48% with ST-Router.

**Effect of sharing weights and role of attentions**    We conducted experiments with two settings. First, STAR-CAPS with separate weights with attention modules. We didn't notice improvement on MNIST. On CIFAR10 $\{32, 8, 8, 8, 10\}$ achieved 91.31% vs. 91.23%; however, the train/test time were significantly higher due to extra matrix multiplications as in EMCaps. Second, STAR-CAPS with separate weights without attentions; the experiments on MNIST/CIFAR10 showed poor performance. In conclusion, the proposed setting of STAR-CAPS provides best results in general, in terms of accuracy and train/test time while preserving capsule properties.

## B.    Pseudo Code of STAR-CAPS

We provide a brief pseudo code for the forward propagation of a $\texttt{ConvCaps}_\ell(k, n_\ell)$ layer in STAR-CAPS architecture, following the notation and the equations presented in Section 3 in the main paper.

---

**Algorithm 1** Forward pass of $\texttt{ConvCaps}_\ell(k, n_\ell)$ layer in STAR-CAPS architecture.

---

**Input:** a set of the input pose matrices $\mathbb{P}_{\ell-1} = \left\{ \mathbf{P}_i \in \mathbb{R}^{p \times p} \mid i \in \{1, \ldots, n_{\ell-1}\} \right\}$ generated by the lower-level capsules in layer $\ell - 1$.

**Output:** a set of output pose matrices $\mathbb{P}_\ell = \left\{ \tilde{\mathbf{P}}_j \in \mathbb{R}^{p \times p} \mid j \in \{1, \ldots, n_\ell\} \right\}$ generated by the higher-level capsules defined in the current layer $\ell$.

**1.** Transform the **input pose**:

**for all** *pose matrix* $\mathbf{P}_i \in \mathbb{R}^{p \times p}$ and *transformation matrix* $\mathbf{W}_i \in \mathbb{R}^{p \times p}$ **do**
$\quad \mathbf{V}_i^{pre} = \mathbf{P}_i \mathbf{W}_i \quad \mid i \in \{1, \ldots, n_{\ell-1}\}$
**end for**

**2.** Estimate **attention matrices** $\mathbf{A}_{ij} \in \mathbb{R}^{p \times p}$ using **Attention Estimators** $\mathcal{T}_{ij}$:

**for all** *pre-votes* $\mathbf{V}_i^{pre} \in \mathbb{R}^{p \times p}$ **do**
$\quad \mathbf{A}_{ij} \leftarrow \mathcal{T}_{ij}(\mathbf{V}_i^{pre}) \quad \mid i \in \{1, \ldots, n_{\ell-1}\}, \ j \in \{1, \ldots, n_\ell\}$
**end for**

**3.** Estimate **routing decisions** $\delta_{ij} \in \{0, 1\}$ using **Straight-Through Routers** $\mathcal{R}_{ij}$:

**for all** *attention matrices* $\mathbf{A}_{ij} \in \mathbb{R}^{p \times p}$ **do**
$\quad \delta_{ij} \leftarrow \mathcal{R}_{ij}(\mathbf{A}_{ij}) \quad \mid i \in \{1, \ldots, n_{\ell-1}\}, \ j \in \{1, \ldots, n_\ell\}$
**end for**

**4.** Calculate the **output pose**:

**for all** $\mathbf{A}_{ij}$ and $\mathbf{V}_i^{pre}$ **do**
$\quad \tilde{\mathbf{A}}_{ij} = \mathbf{A}_{ij} \oslash \sum_{\substack{i=1 \\ \delta_{ij}=1}}^{n_{\ell-1}} \mathbf{A}_{ij} \quad \mid i \in \{1, \ldots, n_{\ell-1}\}, \ j \in \{1, \ldots, n_\ell\}$
$\quad \tilde{\mathbf{P}}_j = \sum_{\substack{i=1 \\ \delta_{ij}=1}}^{n_{\ell-1}} \mathbf{V}_i^{pre} \odot \tilde{\mathbf{A}}_{ij} \quad \mid i \in \{1, \ldots, n_{\ell-1}\}, \ j \in \{1, \ldots, n_\ell\}$
**end for**

---