[Reviews · NeurIPS 2019]

Reviewer 1



The presented routing mechanism differs from routing by agreement in that routing coefficients are not iteratively determined by evaluating agreement of votes, but by computing self-attention scores and binary routing decisions for each combination of input and output capsules. The combined routing procedure seems to be novel, while the individual parts are inspired by self-attention in the transformer and Gumbel-softmax decisions as used in discrete domains like text processing. The paper is technically sound and is very well written, precisely explaining the method and architecture. I feel confident that I could reproduce the method given the provided information. The paper achieves its goal in providing a more efficient substitute for routing by agreement, which allows the architecture to be applied on real-world datasets like ImageNet, as is shown by the experiments. For me, the biggest issue is that I am not fully convinced that the approach still provides similar advantages and properties as capsule networks with routing by agreement. Routing by agreement introduces a very strong inductive bias, assigning high routing coefficients only to connections that are supported by a lot of other connections. In fact, routing by agreement can be interpreted as robust pose estimation, finding the most plausible model and filtering out outliers in supporting evidence through fixed-function, iterative re-weighting. We contribute robustness to changes in pose to this procedure. In the proposed approach, this fixed procedure is eliminated, as routing coefficients only depend on the individual pre-votes and not on the joint behavior of input capsules. The question I am asking myself is: Are these still capsule networks (in the sense of Sabour et al. with routing by agreement), or are these "just" CNNs with multi-head attention? In any case, the experimental results support the effectiveness of the approach when it comes to classification accuracy. However, if it is the case that the properties of capsule networks are not preserved and the method does also not beat other CNN architectures in terms of classification accuracy, I think the results are not significant enough for the paper to be accepted. I therefore ask the authors to provide additional evidence and/or arguments that indicate that the positive properties of capsule properties are preserved by the proposed relaxation (see below). ---- Update: I thank the authors for the extensive response, which addresses most of my issues. I increased my score to 6 since the authors provided evidence for the preservation of capsule network properties on smallNORBS and affNIST. Also, they provided a large ablation study, which is able to compensate weaknesses in motivation by empirical evaluation and brings the work to a level on which it can be accepted. However, at least for me, it is still not clear why exactly the proposed approach works as good as it does. I encourage the authors to further work on the motivational background to improve the takeaway of the reader. A thought: P is called a pose but it is modified by element-wise multiplication instead of linear transformations, thus, ignoring the underlying space of geometric poses and treating it simply as a set of scalar features. This means that the output poses of a layer are able to behave arbitrarily when the input poses change consistently. The network may be able to learn consistent behavior (is it? That seems hard to grasp). However, the forced inductive bias is gone. I find it surprising that it seems to be not necessary for the good generalization to novel viewpoints.

Reviewer 2



This paper replaces the iterative routing (cluster finding) used in EMCapsules with a transformer-like attention. Since there is no notion of cluster finding and it is similar to transformers, the units do not have organic existances and the capsules only have instantiation parameters. Therefore, to sparsify the network authors design a binary decision block which prunes the connections (existance of the connections) in addition to the attention. For the classification, they improvise a notion of capsule existence by summing sigmoids over elements of its instantiation matrix. This is an original work which proposes a network for computer vision with grouped activations. Alike CapsuleNetworks, in each layer there are groups of neurons that cooperate with each other to explain incoming signals. In this work in each layer groups of neurons compete with each other for sending their signals (equation 4, sum over i) where as in Capsule Networks capsules compete with each other for receiving a signal (sum over j). The idea behind Capsule Networks is that each unit has a single parent, therefore the normalization should be over j which automatically sparsifies the network. Because some of the upper capsules may receive 0 signal. In the current proposed work a single capsule in layer L can activate all of the capsules in layer L+1 by sending its vote to all of them. This is an interesting work which is deviates from EMCapsules in several ways. Such as matrix multiplying the weights and poses inspired by affine transformations between part-wholes in EMCapsnet is here replaced with element wise multiply. Expanding on the reasoning behind this change would improve the manuscript. Also finding strong agreements by finding clusters in all bottom groups of EMCapsnet, is here replaced with a pairwise top-down agreement. In the experiments they show that the new proposed model have better accuracy in compare to EMCapsNet and their viewpoint generalization ability does not deteriorate. Expanding on the viewpoint generalization experiments will make this submission stronger. Generalization on rotation of smallNORB and affnist especially. Room for improvement: The importance of each novel component is not explained, or explored. An ablation study which experiments without the straight-through router would clarify the role of the attention module vs the connection sparsifier. Furthermore, an ablation which does normalization on j rather than i (like EMCapsnet) would make StarCaps closer to EMCaps and therefore more comparable. The role of the specific attention module (with several conv layers) vs a simple linear layer is not clear. Line 264-280: is inaccurate. EMCapsnet has two threshold beta_a and beta_u which threshold the activation of a capsule based on std and support (how many inputs are assigned) of the cluster. If std of clusters are larger than beta_u or number of capsules is smaller than beta_a the capsule will deactivate. These are proportional to the std of weights and total number of available points (number of capsules in previous layer). The initialization values in Hinton 2018 are for 32 capsules. Changing the initial value proportional to the change of the number of capsules controls the sparsity of EMCapsnet layers. Also in Table 1, authors can report proportion of active capsules, or proportion of routing factors above .5 after 3 routing iteration for EMCapsNet as well. Currently the vanishing pose argument is only backed by the drop in accuracy which given Fig. 2 is expected anyway. ************************************************* Thank you for providing the ablation study and affnist results. The performance gain with the lack of single parent assumption is interesting. I am increasing my score to 7. I would still urge the authors to provide the smallNORB rotation generalizability as well. Providing the results only on Azimuth makes the submission weaker.

Reviewer 3



Post-Rebuttal: Many thanks to the authors for their rebuttal and the effort they put on addressing some of the issues identified by the reviewers. The results presented seem to be on par with EM but the technique is a novel spin on the capsule routing. I am still not convinced that the main principles of Capsules are represented in the formulations presented but I am curious about the future iterations and improvements. ---------------------------------------------------------------------------------------- Originality: The paper explores an interesting avenue involving capsule networks, by bridging attention logic and capsule convolutional nets. The idea of using attention in capsule networks in itself is not brand new [Zhang et al., 2018, Li et al., 2019], however the formulation presented here is substantially different as it replaces the iterative routing with differentiable binary decision neurons for connecting between lower and higher level capsules via straight-through gumbel-softmax. Quality: The technical content of the paper appears to be generally sound and the overall argument for the approach is persuasive enough. It seems plausible that replacing the iterative/recurrent nature of other routing algorithms with attention can work better, as is the case with recurrent networks and the Transformer. One concern is the use of a global average pooling layer in the Decision-Learner (195-197) and activation probability (174-177) which seems to be against the capsule network philosophy outlined in the introduction and in previous works. I may be missing some details so perhaps further explanation and rationale for this decision could be made explicit. The authors also opted for a differentiable hard attention module as opposed to soft when connecting capsules from a lower level (object parts) to higher level capsules (objects). The goal of capsule networks is to learn part-whole relationships and increase resilience to input transformations. So one can imagine a lower level capsule representing a wheel which should be connected in part to a higher level capsule representing a car, and another representing a different vehicle and the connection strengths depend on the objects properties such as scale/skew/viewpoint etc. Among other examples, it may seem rudimentary to have a purely binary decision on whether to connect a part of an object. Perhaps the authors could comment on this and clarify that their approach does indeed stay true to the original capsule network philosophy/logic. Clarity: The paper is well written and the notation used is clear enough. However, a minor issue is that it may be difficult to follow for readers who are unfamiliar with capsule networks. The authors could consider making the introduction of capsule networks in general a little more detailed or aided by a diagram. Moreover, in light of previous work on capsules the authors could also consider adding a pseudocode Algorithm of their approach to make their explanation clearer and easier to replicate for the community. Significance: The paper explores capsule networks which is a field in its early stages of research so the results presented are not state-of-the-art, which is to be expected, so the comparisons are made with previous work on capsules. There are several comparisons made with the EM routing algorithm on multiple datasets, showing that there is merit to the approach. However, the majority are made using MNIST which is indicative but not the best benchmark to use since it’s overused/simplistic. The novel view-point performance on smallNORB is nicely presented, but an overall classification performance on this dataset is missing. This would be a great addition/comparison to draw since smallNORB is the primary benchmark for capsule networks as EM routing holds the state-of-the-art performance. The authors could also consider comparing directly to other works aside from the original EM routing paper to validate their approach and strengthen their argument. The experiments on CIFAR/ImageNet indicate that the approach is somewhat scalable and that there is room for improvement going forward. In summary, the potential impact of capsule networks in general is significant and this paper proposes an alternative method so it is interesting to the capsule network research community. Zhang, N., Deng, S., Sun, Z., Chen, X., Zhang, W. and Chen, H., 2018. Attention-based capsule networks with dynamic routing for relation extraction. arXiv preprint arXiv:1812.11321. Li, J., Yang, B., Dou, Z.Y., Wang, X., Lyu, M.R. and Tu, Z., 2019. Information Aggregation for Multi-Head Attention with Routing-by-Agreement. arXiv preprint arXiv:1904.03100.

[Author Response · NeurIPS 2019]

We thank the reviewers (**R4**, **R5**, **R6**) for the useful feedback. Below we address their questions.

**(R4, R5, R6) Is STARCAPS a capsule network?** The table to the
right shows the test accuracies of two types of experiments on Small-
NORB (novel, familiar viewpoints). ***Type1:*** 3 runs of models EMCaps
$\{64, 8, 16, 16, 5\}$, STARCAPS $\{32, 8, 8, 8, 5\}$, fully trained on familiar
views and tested on both novel and familiar views. ***Type2:*** EMCaps

|  | Type1 (low capacity models) | | Type2 (high capacity models) | | |
|---|---|---|---|---|---|
| Model #params | EMCaps 68K | STARCAPS 73K | CNN 4.2M | EMCaps 316K | STARCAPS 318K |
| *Familiar* | 95.66±0.03 | 95.72±0.02 | 96.3 | 96.3 | 96.3 |
| *Novel* | 86.12±0.05 | 86.07±0.03 | 80.0 | 86.5 | 86.3 |

$\{32, 32, 32, 32, 5\}$, STARCAPS $\{32, 32, 16, 16, 5\}$, trained on familiar
views and early stopped when test accuracy reached $96.3\%$ (as the CNN model in Table 2 of [EMCaps, Hinton et al]).
In *Type1*, we notice that STARCAPS achieves comparable results (small difference in accuracy) to EMCaps both on
familiar and novel viewpoints. In *Type2*, on the novel viewpoints, STARCAPS performs dramatically better than CNN
model (+6.3%) and its accuracy is only slightly lower than EMCaps (-0.2%).
affNIST results: We trained STARCAPS $\{32, 8, 16, 16, 10\}$ on MNIST following the data augmentation mentioned by
the authors on OpenReview/ICLR18. The test accuracy of STARCAPS on affNIST is 93.03% vs. 93.1% for EMCaps
$\{32, 32, 32, 32, 10\}$. In conclusion, the results show that STARCAPS is capable of detecting novel viewpoints similarly
to EMCaps, retaining capsules properties.

**(R4) Performance of STARCAPS vs. CNNs** Although the main purpose of STARCAPS is to alleviate the computa-
tional complexity of baseline capsule networks while being able to detect viewpoint variations, STARCAPS models
achieve accuracies nearly on par with those modern CNN models. On CIFAR10, STARCAPS: 91.23%, #params=80K
vs. ResNet20: 91.25%, #params=270K vs. ResNet110: 93.57%, #params=1.7M. On CIFAR100, STARCAPS: 67.66%
vs. ResNet38: 68.54% vs. ResNet110: 71.21%. It is possible that scaling up STARCAPS models to match #params in
ResNet, would lead to better performance; however this requires further extensive study.

**Ablation studies** **(a)** *ST-Router*: Removing ST-Router leads to lower performance. On MNIST, STARCAPS model
$\{32, 8, 16, 16, 10\}$ achieves 99.41 w/o ST-Router & 99.59 with ST-Router, while $\{32, 4, 64, 4, 10\}$ achieves 98.37
w/o ST-Router & 99.48 with ST-Router. **(b)** *Single parent assumption*: The single parent assumption enforced in
EMCaps/DynamicCaps, while it may allow better encoding of entity representation, it imposes a limitation as it ignores
actual/natural use cases for object recognition. We tested STARCAPS models designed to force the single parent assump-
tion, the results were comparable to the proposed STARCAPS models on MNIST; however on {CIFAR10; CIFAR100}
the results were inferior due to varied clutter in backgrounds, $\{89.91; 62.33\}$ vs. STARCAPS $=\{91.23; 67.66\}$ &
EMCaps $=\{88.10; n/a\}$. **(c)** *Effect of sharing weights & role of attention*: Experiments with two settings. First, STAR-
CAPS with separate weights with Attentions. We didn't notice improvement on MNIST; on CIFAR10 $\{32, 8, 8, 8, 10\}$
achieved 91.31 vs. 91.23. However, the train/test time were significantly higher due to extra matrix multiplications
as in EMCaps; we couldn't train models on CIFAR100. Second, STARCAPS with separate weights w/o Attentions.
Experiments on MNIST/CIFAR10 showed very poor performance. In conclusion, the proposed setting of STARCAPS
provides best results in general, in terms of accuracy and train/test time while preserving capsule properties.

**(R5) Visualization of instantiations params** We will include visualizations in the supplementary material.

**(R5) Vanishing pose** In EMCaps the initializations of vector parameters $(\beta_a, \beta_u)$ control the initial sparsity. In practice,
according to our experiments, even with careful initialization of parameters, some EMCaps models suffer from unstable
performance due to numerical issues with gradients (vanishing gradients). This was confirmed by the authors of EMCaps
(see OpenReview/ICLR18 comments). We noticed unstable performance in EMCaps when a capsule layer has very
large #capsules compared to lower/higher capsules, and when multiple adjacent layers have very large #capsules. The
routing in STARCAPS automatically prunes the unneeded capsules without being sensitive to #capsules/initializations.
We will update Table 1, fix the argument in lines (264-280), and add clarifications about the instability issues.

**(R6) SmallNORB overall results** STARCAPS: **S1**=$\{32, 8, 8, 8, 5\}$, $73K$, 98.0%; **S2**=$\{32, 32, 16, 16, 5\}$, $318K$,
98.2%; vs. EMCaps: **E1**=$\{64, 8, 16, 16, 5\}$, $68K$, 97.8%, **E2**=$\{32, 32, 32, 32, 5\}$, $316K$, 98.2%

**(R6) Related work, capsules introduction, pseudo code** We will add pseudo code in the supplementary material,
references to (Zhang et al.) and (Yang et al.), and a more detailed introduction to capsule networks.

**(R6) Global avg pooling (GAP)** In STARCAPS, the role of GAP is not routing. We use GAP in Decision-Learner
internally in ST-Router ($\mathcal{R}_{ij}$) to rapidly capture confidence maps from an attention matrix $A_{ij}$. The role of $\mathcal{R}_{ij}$ is to
estimate binary connectivity decision signal between two capsules; each $\mathcal{R}_{ij}$ determines the connectivity between a
single lower-level capsule and a single higher-level capsule (one-to-one), and not routing between lower and higher
capsules (compared to dynamic routing in capsules, or static routing between neurons using max-pooling in CNNs).
The name "ST-Router" may raise confusion with "routing" in EMCaps/DynamicCaps. Each ST-Router acts as a *gating*
mechanism between two capsules, whereas the whole set of ST-Routers acts as a *routing* mechanism between all
capsules. We will change the name to "ST-Gate". We also use GAP after the final output layer (ClassCaps) to calculate
the final activation probabilities from the sigmoid of pose matrix, and not for routing between capsules.

**(R6) Hard attention** STARCAPS uses both soft and hard attention modules. The soft attention (Attn-Estimator)
estimates soft relevance signal for each higher capsule which is used for scaling the pre-vote. To sparsify the network
we use a hard attention module (ST-Router). Each route can be seen as a double-attention (soft & hard) mechanism.

[Meta-Review · NeurIPS 2019]

The reviewers have converged to support this paper, However, some questions raised were not addressed in the response (see the reviews). I strongly recommend the authors to address those to increase the impact of the paper.